# Immunomodulatory Drugs for the Treatment of B Cell Malignancies

**DOI:** 10.3390/ijms22168572

**Published:** 2021-08-09

**Authors:** Nikolaos Ioannou, Khushi Jain, Alan G. Ramsay

**Affiliations:** Faculty of Life Sciences & Medicine, School of Cancer & Pharmaceutical Sciences, King’s College London, London SE1 9RT, UK; nikolaos.1.ioannou@kcl.ac.uk (N.I.); khushi.jain@kcl.ac.uk (K.J.)

**Keywords:** B cell malignancy, immunomodulatory drugs, lenalidomide, CELMoDs, avadomide, CLL, NHL, lymphoma

## Abstract

Accumulating evidence suggests that the tumor microenvironment (TME) is involved in disease progression and drug resistance in B cell malignancies, by supporting tumor growth and facilitating the ability of malignant cells to avoid immune recognition. Immunomodulatory drugs (IMiDs) such as lenalidomide have some direct anti-tumor activity, but critically also target various cellular compartments of the TME including T cells, NK cells, and stromal cells, which interfere with pro-tumor signaling while activating anti-tumor immune responses. Lenalidomide has delivered favorable clinical outcomes as a single-agent, and in combination therapy leads to durable responses in chronic lymphocytic leukemia (CLL) and several non-Hodgkin lymphomas (NHLs) including follicular lymphoma (FL), diffuse large B cell lymphoma (DLBCL), and mantle cell lymphoma (MCL). Recently, avadomide, a next generation cereblon E3 ligase modulator (CELMoD), has shown potent anti-tumor and TME immunomodulatory effects, as well as promising clinical efficacy in DLBCL. This review describes how the pleiotropic effects of IMiDs and CELMoDs could make them excellent candidates for combination therapy in the immuno-oncology era—a concept supported by preclinical data, as well as the recent approval of lenalidomide in combination with rituximab for the treatment of relapsed/refractory (R/R) FL.

## 1. Introduction

Despite recent advances in diagnosis and treatment, B cell malignancies still account for significant morbidity and mortality worldwide. The introduction of the anti-CD20 monoclonal antibody (mAb) rituximab more than 20 years ago, has contributed to great clinical benefit in both a front-line and a R/R setting as a single-agent and in combination with other therapeutic strategies in CLL and B cell NHL [1]. However, despite the improvements in progression free survival (PFS) and overall survival (OS), a significant number of patients either fail to respond or eventually relapse [2]. Management of CLL and B cell NHL, especially in the R/R setting or for high-risk patients, highlights major treatment challenges, particularly in patients with chemoimmunotherapy-resistant disease.

Numerous novel molecularly-based targeted treatments have been developed and investigated in the last two decades, with some of them exhibiting very favorable clinical outcomes, such as B cell receptor (BCR) signalling targeting tyrosine kinase inhibitors (e.g., ibrutinib and idelalisib) [3]. Ibrutinib, an irreversible inhibitor of Bruton tyrosine kinase (BTK) has shown profound benefits in high-risk CLL patients, with durable response rates and prolonged PFS and OS in a R/R CLL setting however, the majority of responses are partial. In addition, continuous treatment is required for disease control [4,5]. More recently, BCL2 has also been utilized as a therapeutic target in CLL. Venetoclax is a BCL2 antagonist that has shown high response rates with deep remissions in front-line and R/R CLL settings in combination with anti-CD20 antibodies [6,7]. Despite these advances, there are still challenges associated with treatment with small molecule inhibitors, which include toxicity and the risk of developing drug resistance [8]. Thus, it has become pertinent to develop combinational therapeutic strategies utilizing agents with distinct mechanisms of action.

Immunotherapy is rapidly emerging as a major treatment option in first-line and R/R setting in B cell malignancies [9]. Immunomodulatory drugs (IMiDS) such as lenalidomide (Revlimid) and pomalidomide are thalidomide derivatives, with pleiotropic effects in human malignancies. These agents exhibit a plethora of anti-cancer properties including anti-angiogenic, anti-proliferative and immunomodulatory effects that have demonstrated significant clinical activity for the treatment of both untreated and R/R B cell hematological malignancies as a single-agent and in combination with other agents including rituximab [10,11,12,13,14,15].

Lenalidomide is an orally active immunomodulatory drug and was the first compound of this group of immunomodulatory agents to receive US Food and Drug Administration (FDA) approval, following more than a decade of clinical investigations. Lenalidomide was initially approved in 2005, for the treatment of patients with myelodysplastic syndromes (MDS)-associated anaemia, and in 2006, for the treatment of relapsed/refractory multiple myeloma (MM) in combination with dexamethasone [16]. In 2013, lenalidomide was also approved for the treatment of relapsed (following two prior therapies) MCL, while in the same year pomalidomide was approved for the treatment of R/R MM. Notably, in 2019, the USFDA (and the EMA) approved this agent for the treatment of previously treated FL and marginal zone lymphoma (MZL) in combination with rituximab, based on the results of two phase III clinical trials, while in 2020, the combination of lenalidomide with the cytolytic CD19 targeting monoclonal antibody tafasitamab received accelerated approval for the treatment of R/R diffuse large B cell lymphoma (DLBCL) patients [17,18,19,20,21]. More recently, avadomide (CC-122) a next generation CELMoD, has demonstrated promising clinical activity in R/R DLBCL patients as a single-agent. Avadomide has also been well-tolerated in combination therapy with anti-CD20 monoclonal antibodies including obinutuzumab [22]. Avadomide exhibits its antitumor effects by binding to the same protein target, cereblon (CRBN), as IMiDs, however, preclinical and clinical data suggest that its direct cell autonomous activity (anti-proliferative and direct tumor cytotoxicity), as well as its immunostimulatory effects are superior to that of lenalidomide [23,24].

IMiDs and CELMoDs have a unique mechanism of action (MOA), exerting their anti-malignant effect not only by targeting the tumor cells but also by modulating several non-malignant components of the TME such as T cells, NK cells, tumor-associated macrophages (TAMs, also known as ‘nurse-like cells’, [NLCs]) and dendritic cells (DCs), which are believed to play an important role in lymphoma progression and survival. The unique attributes of these compounds renders them potentially excellent candidates for future combinational therapeutic strategies for improving survival and a clinical benefit in these malignancies [25].

In this review, we discuss the mechanisms of action of both IMiDs and CELMoDs in CLL and B cell NHL and focus on their effects on cellular components within the TME—their immunomodulatory activity towards NK cells, T cells and stromal cells. In addition, the clinical efficacy of the IMiD lenalidomide and the CELMoD avadomide as single-agents and in combination with other therapies is briefly summarized. Lastly, we discuss the potential future clinical development of these agents as combination partners.

## 2. CRBN as a Target for IMiDs and CELMoDs

CRBN, a ubiquitously expressed substrate receptor protein of the Cullin 4 RING E3 ubiquitin ligase complex (CRL4CRBN), has been identified as the primary target for IMiDs and CELMoDs. Immunomodulatory agents contain a conserved glutarimide moiety that directly binds with a tri-tryptophan hydrophobic pocket in the cereblon binding domain, activating the E3 ligase activity of CRL4CRBN, modulating substrate specificity and preventing autoubiquitylation [23,26,27]. In CLL and NHL cells, IMiDs and CELMoDs re-target cereblon-dependent activity by promoting the recruitment and its selective binding to two haematopoietic transcription factors Ikaros (IKZF1) and Aiolos (IKZF3), resulting in their ubiquitination and subsequent proteasomal degradation. Ikaros and Aiolos are zinc finger proteins that act as transcriptional regulators of both B and T cell development [28]. Their reduced abundance in tumor B cells elicits anti-neoplastic and anti-proliferative effects. Importantly, Ikaros and Aiolos are also known repressors of interleukin-2 (IL-2) transcription in T cells, and their downregulation leads to an increase in IL-2 production which contributes in part to the immunostimulatory effects of IMiDs and CELMoDs. [23,26,29,30,31].

## 3. Direct Effects of IMiDs on Malignant B Cells

IMiDs have demonstrated direct anti-neoplastic activity against malignant B cells in specific tumor types, including CLL and NHL (DLBCL, FL and MCL). Despite the lack of a direct cytotoxic effect on malignant CLL cells, IMiDs directly affect them via several mechanisms including reduction of their proliferation rate, modification of their adhesion and migration properties and an increase of their recognition by the immune system [23,32]. Lenalidomide downregulates IRF4 and MYC levels and enhances expression of p21WAF/Cip1, a regulator of cyclin-dependent kinases (CDKs), which leads to cell cycle arrest in the G0-G1 phase on malignant B cells [33]. Furthermore, lenalidomide treatment of CLL cells leads to an upregulation of several co-stimulatory molecules such as CD80 and CD86, and subsequently their recognition by and activation of the immune system while it impairs CLL migration via upregulation of CORO1B (Coronin, Actin binding protein) and reduction of RhoH expression, thereby impeding pro-survival CLL cell−microenvironment interactions [30,34,35,36,37].

Lenalidomide also exhibits in vitro anti-tumor activity on several NHL tumor types including DLBCL, FL and MCL [38,39,40,41]. Lenalidomide was shown to increase the percentage of malignant B cells (lymphoma cell line) arrested in the G0-G1 phase by upregulating p21WAF/Cip1 and the inhibition of Akt and GRB-2 associated binding protein 1 (GAB1) phosphorylation [42]. Notably, lenalidomide has been shown to have higher efficacy against the activated B cell (ABC)-like subtype of DLBCL cell compared to germinal center B cell (GCB)-like DLBCL, where it preferentially inhibits cell proliferation and reduces tumor growth. In the ABC-subtype of DLBCL, lenalidomide leads to cell cycle arrest and/or apoptosis via downregulation of IRF4 and SPIB transcription factors and by the induction of a type I interferon response (interferon β, IFNB) in a cereblon-dependent manner [40,41]. In addition, lenalidomide has been shown to downregulate the expression of immune checkpoint molecules such as PD-L1 in lymphoma and myeloma cells leading to increased formation of lytic immune synapses and anti-tumor cytotoxicity by NK and CD8^+^ T cells [43]. Lenalidomide has exhibited anti-tumor activity in MCL cell lines as well, while it has been shown to enhance the expression of genes involved in immune responses including CD40, CD58, CD86 and CD1c [44,45].

Despite IMiDs and CELMoDs sharing a common target, CRBN, several preclinical and clinical studies have demonstrated differential activity of lenalidomide and avadomide in DLBCL cells. Interestingly, while lenalidomide has exhibited anti-tumor activity against ABC-DLBCL cell lines only, avadomide has shown direct apoptotic activity against both GCB- and ABC-DLBCL subtypes, attributed to enhanced transcription of interferon (IFN)–stimulated genes and associated anti-tumor activity but independent of IFN-α, -β and -y production and secretion by malignant B cells [23]. In addition to significantly higher reductions in the expression levels of Aiolos and Ikaros, avadomide treatment was also found to uniquely increase the expression of transcription factors IRF7 and DDX58, suggesting that these two agents may interact with distinct and unique substrates compared with lenalidomide. In addition, lenalidomide interacts with a specific substrate called casein kinase 1α, which is not affected by avadomide, further supporting this concept [23].

## 4. IMiDs and the TME

Recent findings have revealed the importance of the interplay between tumor and non-malignant cells such as T cells, NK cells, DCs, macrophages, stromal cells and endothelial cells (ECs), in the lymph nodes and the bone marrow which constitute the TME. Several studies have shown that this active crosstalk is critical for disease establishment and progression through the support of tumor survival, growth and migration, as well as by facilitating immune evasion of tumor cells [46,47].

Initial evidence on the critical role of TME came by several studies demonstrating that ex vivo co-culture of CLL cells with non-malignant TME cells, including bone marrow-derived MSCs (mesenchymal stem cells) and monocyte-derived NLCs, improved cell survival when compared to CLL cells cultured alone [48,49]. In addition, NLCs provide an important immunoregulatory role in CLL, by attracting other immune cells (e.g., T cells) through the production of chemokines CCL3 and CCL4 [50,51]. Similarly, signals from stromal cells have been shown to support growth and survival of tumor cells in several types of B cell NHL including FL, MCL and DLBCL [52].

Another mechanism by which the TME promotes B cell survival and growth is by the continuous activation of BCR signalling which is required for B cell survival [53]. BCR stimulation leads to the activation of major cell survival signalling pathways including the phosphatidylinositol 3 kinase (PI3K)-AKT pathway and nuclear factor-κB, and it has been shown that its activation can result not only through the interaction of B cells with self-antigens but also through an antigen-independent mechanism [52,54].

In addition to the direct effects on tumor cell survival and proliferation, components of the TME can also facilitate the ability of malignant B cells to avoid recognition and destruction by the immune system. Several mechanisms have been identified so far, generally including changes in the expression levels of molecules that are important for interactions with immune cells (e.g., loss or downregulation of MHC I and II expression), direct immunosuppressive effects by malignant cells through cell−cell interactions (e.g., induction of defective immune synapse formation in T cells by FL and CLL cells) and the recruitment of immunosuppressive cells (e.g., enrichment of tumor sites with regulatory T cells (Tregs) and tumor-associated macrophages) (Figure 1) [43,52,55,56].

### 4.1. Effect on T Cells

The ability of malignant cells to affect the TME leads to a progressive immune suppression that is a hallmark of both CLL and NHL. Through a multitude of mechanisms, such as chronic tumor-associated antigen exposure, increased expression of inhibitory ligands, production of immunosuppressive cytokines and downregulation of co-stimulatory molecules, among others, tumor cells can directly supress the function of T cells [52]. Tumor-infiltrating CD8^+^ and CD4^+^ T cells in CLL and B cell NHL exhibit an abnormal gene expression profile reflecting an immunosuppressed/exhausted phenotype, involving deregulation of various signalling pathway genes including actin cytoskeleton trafficking and cytotoxicity compared to healthy donor T cells. Additionally, the level of Tregs is increased in CLL patients, likely contributing to immune suppression and disease progression due to reduced immunostimulatory cytokine production and activation of effector T cells [57,58,59,60].

Treatment with IMiDs can lead to increased T cell activation and proliferation, by upregulating the release of pro-inflammatory cytokines (e.g., IFN-γ, TNF-α and IL-2) and downregulation of exhaustion-associated marker PD-1, increasing the number of functional CD8^+^ and CD4^+^ T cells and promoting an increased ratio of Th1 versus Th2 subsets in CLL [25,61,62]. In addition, lenalidomide has been shown to decrease the number of Tregs, whilst increasing the number of Th17 cells to a level equivalent to that in healthy subjects [63]. Interestingly, a study by Lee at el. investigating the immunomodulatory activity of lenalidomide, showed a normalization of the numbers of all functional T cell subsets revealing an additional level of T cell modulation [64].

Lenalidomide treatment can also repair immune synapse formation defects in CLL and NHL by enhancing T cell actin polymerization and increasing the function and assembly of several signalling molecules such as WASp, CDC42 (RHO GTPase) and PKC-θ to the synapse, leading to effective lytic formation and activity [43,60]. In addition, lenalidomide has been shown to downregulate the expression of inhibitory checkpoint ligands (CD200, CD270, PD-L1, and CD276) in allogenic co-culture assays, restoring normal Rho GTPase signalling (RhoA, Rac1, Cdc42) and stimulating expression of high-affinity LFA-1 for promoting T cell adhesion and motility function [65].

Preclinical and early-phase clinical studies have shown that avadomide is significantly more potent at promoting the activation of T cell synapses in CLL and NHL and enhancing immune-mediated tumor cell killing compared to the IMiD lenalidomide. Treatment with avadomide increased the number of infiltrating CD4^+^ and CD8^+^ T cells that recognize malignant B cells and enhanced T cell F-actin immune synapse formation, while increasing the number of memory T cells and activated CD8^+^ and NK cells [62,66,67]. Most importantly, avadomide was shown to create an inflammatory T cell secretome by inducing type I and II IFN signalling in patient T cells that stimulates effector proliferation function, T cell motility and the secretion of various pro-inflammatory (TNF-α, IL-2) and chemotactic cytokines (CXCL10, CCL5) that augment the recruitment and activation of granzyme^+^ cytotoxic CD8^+^ T cells. Notably, avadomide upregulated the expression of PD-L1 in the immune TME, that is associated with ‘hot’ inflammatory tumors and sensitive to anti-PD-1/PD-L1 axis immune checkpoint blockade therapy [62].

### 4.2. Effect on NK Cells

NK cells are very important contributors to the innate immunity and have a significant role in immune defense against tumors in CLL and B cell NHL patients, particulary when anti-CD20 antibodies are utilized within therapeutic regimes [68]. Several studies showed that treatment with lenalidomide and the CELMoD avadomide have potent anti-tumoral immunomodulatory effects toward NK cells [34,66,69,70,71].

Lenalidomide has been shown to increase NK cell proliferation and activation, and augment NK cell-mediated cytotoxicity including antibody-dependent cellular cytotoxicity (ADCC) against tumor cells and restore immune synapse formation [66,72,73,74,75]. IMiDS can increase the number of activated NK cells through the upregulation of co-stimulatory surface molecules such as CD80, CD83 and CD86 and the restoration of NKG2D activating ligand receptor on CLL NK cells as well as the upregulation of natural cytotoxicity receptor NKp30 in both CLL and FL patients [74,75]. It has also been suggested that increased NK cytotoxicity may result from an indirect effect of lenalidomide treatment via the elevated production of IL-2 by T cells [70]. In MCL patients, cytotoxic NK cells (CD3^−^ CD56^+^ CD16^+^) subset substantially expanded following lenalidomide therapy, which not only stimulated tumor-cell killing, but was also associated with longer PFS and OS rates [76]. A study by Guiliani et al. demonstrated that lenalidomide is able to downregulate the expression of immune checkpoint molecule PD-1 on NK cells, aiding in the restoration of NK-mediated cytotoxicity [72]. In addition, it has been shown that lenalidomide can increase NK- and monocyte-mediated ADCC against NHL cell lines treated with rituximab. This effect was not only dependent on the binding of antibody (rituximab) to Fc-𝑦 receptors, but also on the presence of drug-induced IL-2 or IL-12 secretion [77]. Similarly, avadomide has shown potent immunomodulatory activity with enhancement of NK cell proliferation and activation by secretion of cytokines such as IL-2 and the expression of molecules enhancing cytotoxicity such as granzyme B and NKG2D receptor [66,67,78].

### 4.3. Effect on Endothelial and Stromal Cells

Vasculogenesis and angiogenesis are crucial processes for the progression of cancer, as they are vital for sustaining tumor growth by ensuring sufficient delivery of oxygen and nutrients to tumor sites. CLL cells can initiate neovascularization via the secretion of pro-angiogenic signals including vascular endothelial growth factor (VEGF), basic fibroblast growth factor (bFGF) and angiopoietin-2 (Ang2), and inducing PKC-βII expression in endothelial cells which has been shown to be essential for the formation of new blood vessels [79,80,81,82]. Similarly, despite the variation of microvessel density among the different types of NHLs, several studies have demonstrated elevated levels of VEGF in certain types of B cell lymphomas such as DLBCL and MCL [83,84]. ECs have been shown to support malignant B cell survival in CLL and NHL through several mechanisms such as production of BAFF and APRIL (CLL) and inhibition of helper CD4^+^ T cells and Th1 polarization (DLBCL) [85,86].

IMiDs such as lenalidomide have been shown to promote their anti-angiogenic effects by inhibiting the production of VEGF and bFGF and block the pro-survival crosstalk between endothelial and tumor cells, thus, inhibiting tumor growth [87]. In particular, levels of the pro-angiogenic factors were downregulated following lenalidomide treatment, especially in patients who responded well to lenalidomide therapy [87]. This is further supported by a study by Reddy et al., which showcased reduced microvessel density and decreasing angiogenesis [88]. In addition, Song et al. showed that lenalidomide can inhibit lymphangiogenesis via depletion of monocytes and macrophages in MCL mouse models [89].

Numerous studies have demonstrated the supportive role of stromal cells on the survival of malignant B cells in CLL and several types of NHL [55]. MSCs, NLCs, follicular dendritic cells (FDCs) and follicular helper CD4^+^ cells (TFH) have been shown to exert their pro-tumor role through both contact-dependent and -independent mechanisms. Lenalidomide can inhibit SDF-1 (CXCL12) production by MSCs indicating the potential of interfering with the proposed CXCL12/CXCR4 migratory axis in NHL [90]. In addition, since lenalidomide can inhibit IL-6 production, it could potentially inhibit the pro-survival activity of bone marrow-derived stromal cells in MCL via subsequent inhibition of IL-6- mediated STAT-3 signalling [91].

Even though the anti-angiogenic effects of avadomide in CLL and B cell NHL have not fully been studied, in vitro studies suggest that avadomide exhibits its anti-angiogenic effects by inhibiting EC migration and invasion, as well as new vessel growth formation. In addition, preclinical studies also suggest that avadomide has greater anti-angiogenic effects compared to the IMiDs lenalidomide and pomalidomide [24,78]

### 4.4. Effect on DCs

DCs are considered to be the most important antigen presenting cells (APCs) and major messengers between the adaptive and the innate immune system. Through the processing and presentation of antigens on their cell surface, they can activate both CD4^+^ and CD8^+^ T cells and thus promote rejection of tumor cells [92]. Lenalidomide and other IMiDs like pomalidomide are able to enhance T cell cross-priming by upregulating the expression of CD86 and MHC class I molecules on DCs in lymphoid organs. This increases the efficacy of antigen-specific presentation (MHC Class I) on naïve CD8^+^ T cells, leading to increased activation and proliferation of CD8^+^ T cells, allowing for better tumor surveillance [93]. Interestingly, in the same study, in addition to MCH class I, pomalidomide was also shown to increase the expression of MHC class II molecules on DCs, which enhanced priming and proliferation of CD4^+^ T cells [93]. Interestingly, avadomide treatment has been shown to enhance DC trafficking to the lymphoid TME in DLBCL patients [66].

## 5. Clinical Efficacy of IMiDs and CELMoDs in CLL and B Cell NHL

Several clinical studies (phase II and III) have revealed promising results for lenalidomide therapy, when used alone or in combination with other agents in CLL and several types of B cell NHL including DLBCL, FL and MCL in R/R and front-line settings (select trials summarized within Table 1). More recently, the CELMoD avadomide has also demonstrated clinical efficacy in R/R B cell NHL including DLBCL, as a single-agent and in combination therapy with anti-CD20 antibodies including obinutuzumab (Table 1).

### 5.1. Lenalidomide Therapy in B-Cell NHL

Lenalidomide treatment of numerous subtypes of B cell NHL in a clinical setting led to durable responses as monotherapy, in patients with R/R, indolent and aggressive disease (DLBCL, MCL), achieving considerably high efficacy in terms of overall response rate (ORR) (Table 1) [17,18,94,95,96,97]. Notably, a large international study (phase II) investigating lenalidomide monotherapy for aggressive lymphomas (217 patients with R/R NHL, DLBCL, MCL, and FL grade III) reported that single-agent lenalidomide was well-tolerated with an overall ORR of 35% (77/217) and 13% (29/217) CR (NHL-003) [95]. Interestingly, a retrospective evaluation of lenalidomide treatment in DLBCL found that the cell of origin appeared to be of great importance, with lenalidomide demonstrating preferential activity in non-GCB (ABC)-DLBCL compared to the GCB subtype, leading to an ORR of 53% in ABC-like DLBCL compared to only 9% in GCB-DLBCL patients. It should be noted that despite the significant differences in ORR, there was no difference in overall survival (OS) between these two patient subpopulations [98]. In a more recent phase II/III trial (R/R, randomized, multicenter), ORRs were similar between these two subpopulations based on subtyping by IHC (immunohistochemistry), but when GEP classification was used for distinguishing ABC and GCB subtype patients, a significant clinical outcome difference was demonstrated (ORR of 45.5% for ABC-DLBCL vs. 21.4% for GCB-DLBCL) [99].

Lenalidomide monotherapy has also been investigated in MCL, which despite the good response rates, has been characterized by relatively short remissions. In an international phase II single-arm, clinical trial (EMERGE- MCL-001) with 134 patients with R/R MCL, following treatment with bortezomib (a proteasome inhibitor), lenalidomide treatment resulted in a significant clinical benefit leading to a 28% ORR (8% CR) [18]. These results, in addition to previous phase II studies, led to subsequent FDA approval of lenalidomide monotherapy in 2013 for the treatment of R/R MCL patients following two previous therapies, one of which included bortezomib [17,18,96,100].

The combination of lenalidomide plus rituximab (R^2^), supported by results of preclinical studies exhibiting enhanced anti-tumor activity, has been extensively studied in several types of B cell NHL and has been associated with very promising results in both front-line and R/R settings, especially in FL patients [12,19,101,102,103,104,105]. R^2^ therapy in previously untreated patients with aggressive or indolent NHLs in phase II/III trials has demonstrated high ORRs with an impressive 95% response rate (CR 72%) in FL patients with long PFS rates of 60 months, compared to lenalidomide monotherapy, highlighting the power of this combination therapy [12,19,101,102,103,104]. In the CALGB randomized (Alliance) trial, a 76% ORR was attained in R/R indolent FL patients receiving R^2^ therapy (CR 39%) versus 53% ORR in in the lenalidomide monotherapy arm (CR 20%). However, grade 3 or 4 toxicity occurred in 52% of the patients in the combination arm with common adverse events including neutropenia, fatigue and rash [104]. In a more recent phase III trial (AUGMENT) of R^2^ immunotherapy in R/R FL and MZL patients, an ORR of 78% was achieved. Even though many patients experienced an adverse event (63%), a longer PFS of 39.4 months was attained [19]. Based on data from several studies, it is evident that R^2^ therapy significantly improves response rates with longer PFS rates but is associated with higher toxicity compared with lenalidomide monotherapy. Nevertheless, the favorable outcomes achieved by this combination treatment in untreated patients has encouraged research interest in developing chemotherapy-free combination therapies.

In addition to R^2^ therapy, lenalidomide has been investigated as part of alternative combinations in the clinic. For example, the combination of lenalidomide with rituximab, cyclophosphamide, doxorubicin, oncovin and prednisone (Len + R-CHOP) as a front-line therapy has revealed promising ORRs (up to 98% in FL) and PFS in both FL and DLBCL patients [11,106,107]. Another combination that has achieved high response rates is R^2^ + ibrutinib with an ORR of 83%, with almost half of the patients exhibiting a complete response (41%) [108]. Overall, clinical studies with lenalidomide therapy have shown that high response rates and CRs with long PFS can be attained in patients in both front-line and R/R settings, especially when combined with rituximab.

### 5.2. Lenalidomide Therapy in CLL

Results from several phase II clinical trials showed that lenalidomide monotherapy, when used as a first-line therapy, led to up to 72% ORRs, while the response rate was considerably lower (up to 32%) in a R/R setting (Table 1). It should be noted that in the majority of cases, the CRs recorded were relatively limited (<20%). The efficacy of lenalidomide in a combinational CLL setting has also been investigated, with other agents such as monoclonal antibodies including rituximab or bendamustine, with ORRs reported up to 95% [109]. Interestingly, another combination that has shown high response rates is lenalidomide plus ofatumumab (anti-CD20 monoclonal antibody) in a R/R CLL setting, with a long PFS of 16 months and 71% ORR [110].

### 5.3. Avadomide Therapy in B-Cell NHL

Early-phase 1 clinical trials have provided evidence that avadomide is an active anti-tumor drug with a manageable safety and tolerability profile. Preclinical and clinical data suggests that avadomide exerts its anti-lymphoma effects in both GCB- and ABC-DLBCL subset types, unlike lenalidomide [23,66]. Avadomide treatment as a single-agent in R/R NHL patients achieved an ORR of up to 60% with the most common adverse events noted being neutropenia, thrombocytopenia, diarrhoea and fatigue [22,24,66]. In previously treated NHL patients, avadomide in combination with obinutuzumab (anti-CD20 monoclonal antibody) has demonstrated promising preliminary therapeutic activity with an ORR rate of 68%, including a CR of 34% [22]. Due to the potent anti-tumor effects exhibited by avadomide against NHL cells, further investigation of this CELMoD in larger studies in combination with novel anti-CD20 antibodies and novel immunotherapies for R/R patients has been proposed.

## 6. Biomarkers of Response to Treatment with LEN/IMiDs

Clinical studies have identified potential biomarkers of response to IMiDs to help predict safety, tolerability and prognosis, with ongoing studies in NHL including DLBCL. Despite CRBN’s established central role in the mode of action of IMiDs, its potential as a biomarker for predicting response to therapy with these immunomodulatory agents presents challenges. CRBN’s ubiquitous expression in combination with its multiple splice variants and lack of correlation between messenger RNA and protein levels renders its assessment a rather challenging task. CRBN-associated transcription factors, Ikaros and Aiolos, may serve as more effective biomarkers for T cell activation, as proof of concept studies have demonstrated decreased Aiolos levels in peripheral T cells following lenalidomide therapy in healthy subjects and patients [31]. Interestingly, Aiolos has demonstrated utility as a pharmacodynamic biomarker following avadomide treatment in NHL patients, however, no correlation between the extent of Aiolos degradation or IL-2 production and clinical response was observed [24]. In MCL patients, Ki-67 has been recognized as a potential biomarker of response treated with chemotherapy, rituximab and bortezomib (inverse association between survival and Ki-67 expression). However, lenalidomide has been shown to be active regardless of levels of Ki-67, hence, further research is required in identifying biomarkers for response to IMiDs [111].

## 7. Future Perspective—New Combinations

Although IMiDs and CELMoDs have demonstrated clinical activity against CLL and NHL, it is clear that combination therapy will be required to improve response rates, as well as depth of response. Immunotherapy has rapidly become an important part of the treatment of haematological and solid malignancies. These therapies include cell therapies (adoptive cell therapies (ACT), chimeric antigen receptor (CAR)-T cell therapy), bispecific antibodies (T cell engagers), and monoclonal antibodies (including immune checkpoint blockade). As with IMiDs and CELMoDs, many of these immunotherapies have demonstrated promising therapeutic activity as single-agents, but the development of combination therapy is pertinent for improving survival outcomes in patients and working toward curative therapy. Due to their pluripotent effects, including direct anti-tumor effects on malignant B cells, as well as their ability to activate cytolytic immune cells, IMiDs and CELMoDs have the potential to complement other therapies including immunotherapy. The potential to enhance anti-tumor immune responses by overcoming an immunosuppressive TME makes them excellent candidates for combinational immunotherapies, providing toxicity can be managed (Figure 2).

### 7.1. Small Molecule Inhibitors

Given the established potent activity of several molecular targeted therapies such as BTK, PI3K and proteasome inhibitors, the potential synergy of these drugs when paired with IMiDs/CELMoDs could further improve survival outcomes and efficacy. Several preclinical studies appear to support this notion. For example, the combination of lenalidomide with bortezomib led to a synergistic effect against FL and MCL cell lines in vitro, while the addition of lenalidomide to ibrutinib treatment led to enhanced down regulation of IRF4 and apoptosis of DLBCL cell lines and in particular those of the ABC subtype [41,112]. However, results of subsequent clinical trials, where lenalidomide treatment was evaluated in combination with rituximab and ibrutinib or idelalisib, showed high toxicity in relapsed or refractory CLL and indolent lymphoma [113,114]. Similarly, despite the promicing efficacy, the combination of venetoclax, a BCL-2 inhibitor with lenalidomide and rituximab in patients with R/R lymphomas was associated with a high degree of neutropenia [115,116]. In addition to the ftoxicity profiles, the therapeutic potential of the combination of targeted therapies such as ibrutinib or idelalisib with IMiDS may be hampered by the altered immunomodulatory effects of combining these agents. For example, in a preclinical study investigating the combination of lenalidomide with idelalisib (a PI3Kδ inhibitor), PI3Kδ inhibition resulted in the abrogation of some of the positive immunomodulatory effects of the IMiD and in particular the expression of co-stimulatory molecules CD80 and CD86 on CLL B cells [117]. Furthermore, ibrutinib was shown to inhibit ADCC induced by anti-CD20 antibodies rituximab or obinutuzumab aganist MCL cell lines which could not be overcome by the addition of lenalidomide [118]. Despite these limitations, several clinical trials have shown significant efficacy from the chemotherapy-free combination of ibrutinib, rituximab and lenalidomide in DLBCL and MCL, indicating that careful study design, toxicity management and in vivo drug activity could allow such combination therapies to benefit NHL patients, especially in a R/R setting where very few viable treatment options exist [119,120,121].

### 7.2. Monoclonal Antibodies

The clinical success of established combinations in CLL and B cell NHLs, namely anti-CD20 (rituximab and obinutuzumab) antibodies with IMIDS, further supports the utilization of IMiDs/CELMoDs in such a setting. Several trials investigating the potential of combination therapy of lenalidomide with novel tumor-targeting monoclonal antibodies are currently underway with some initial data showing significant efficacy. For example, polatuzumab vedotin, an antibody−drug conjugate that is an anti-CD79b (a BCR component) antibody covalently conjugated to an anti-mitotic cytotoxic agent has demonstrated promising clinical efficacy in combination with obinutuzumab and lenalidomide in patients with R/R FL [122,123]. In addition, an anti-CD19 monoclonal antibody MOR-28 (Tafasitamab), in combination with lenalidomide in a R/R DLBCL setting, has shown significant clinical benefits with durable response rates [21].

### 7.3. Immune Checkpoint Blockade, ICB Antibodies

Early-phase clinical trial results have demonstrated the significant efficacy of ICB and, in particular, anti-PD-1 therapy, in specific lymphoma subtypes such as classical Hodgkin lymphoma (cHL) and primary mediastinal B cell lymphoma [9,56,124]. However, clinical results for anti-PD-1 monotherapy in NHLs including DLBCL and FL have been disappointing, with similar findings in R/R CLL, indicating that ICB monotherapy is unable to overcome profound T cell tolerance (and exhaustion) within the TME in these B cell lymphomas [9,62,125,126]. HLs have a high expression of PD-L1 in the TME, which not only plays a significant role in the pathogenesis of these lymphomas (by driving an ineffective host anti-tumor immune response) but has also advantageously been correlated to response to ICB therapy [9,124,125,126]. Generally, T cell-inflamed tumors have been shown to respond well to ICB therapy and in addition to high levels of expression of PD-L1, are also characterized by high immune cell infiltration and gene expression profiles enriched for interferon-γ (IFN-γ) signaling. The lack of clinical activity by PD-1 blockade in NHLs and CLL has highlighted the importance of developing novel powerful combinations with other therapeutic drugs which can overcome the non-inflamed phenotype of their immune TME landscape. Notably, preclinical data has shown that treatment with the CELMoD avadomide can reprogram non-inflamed CLL and NHL TMEs into inflamed ones, enabling response to checkpoint blockade therapy, as a result of type I and II IFN-induced inflammatory signaling in endogenous T cells that triggers proliferation, motility and lytic activity. Further studies with this combination will have to ensure that toxicity is managed to realize the therapeutic anti-tumor potential in the clinic [62].

In addition to PD-1 and CTLA-4, novel immune checkpoint molecules are emerging, such as the lymphocyte activation gene-3 (LAG-3), T cell immunoglobulin and mucin-domain containing-3 (TIM-3) and T cell immunoglobulin and ITIM domain (TIGIT), among others, the therapeutic potential of which is currently being evaluated in preclinical studies and clinical trials in several types of cancer, expanding the repertoire for future combinational immunotherapies [127].

### 7.4. Bispecific Antibodies

A novel emerging type of immunotherapy agents are bispecific antibodies (BsAbs) such as bispecific T cell engagers (BiTEs). BiTE antibodies have dual specificity for a tumor antigen and the T cell CD3 receptor and function by retargeting cytotoxic T cells to tumor cells leading to lysis of the latter, a therapeutic approach appealing for the treatment of B cell malignancies [128]. Blinatumomab, a CD19/CD3 BiTE that has been FDA approved for the treatment of R/R B cell precursor acute lymphoblastic leukemia (B-ALL), has demonstrated promising clinical efficacy in R/R DLBCL patients, achieving ORRs of up to 89% [129,130,131]. Blinatumomab is currently being evaluated in phase I studies in R/R NHL patients in combination with IMiD lenalidomide (NCT02568552) and preliminary results suggest that this combination demonstrates encouraging efficacy and has been well-tolerated [132].

Similarly, preclinical and preliminary clinical studies of CD19-CD3 BsAbs or novel BiTEs that target CD20 and CD3, such as glofitamab CD20-TCB (RG6026), odronextamab (REGN1979) and mosunetuzumab, have exhibited high potency and a significantly long half-life enabling a safer administration approach, as well as demonstrated clinical efficacy in R/R NHL [133,134,135,136,137]. Mosunetuzumab and CD20-TCB are currently being investigated in combination with lenalidomide in early-phase I clinical trials in R/R FL patients (NCT04246086) [138,139].

### 7.5. CAR-T Cell Therapy

Chimeric antigen receptor (CAR)-T cell therapy has rapidly emerged as a highly promising immunotherapy, having achieved remarkable results for the treatment of B cell malignancies. In this personalized therapy, autologous T cells are genetically engineered to produce an antigen receptor that specifically binds to surface markers of cancer cells, triggering an anti-tumor immune response [140]. CD19-directed CAR-T cells, including axicabtagene ciloleucel, tisagenlecleucel and lisocabtagene maraleucel, have demonstrated potent clinical efficacy, achieving durable responses and improving patient survival outcomes, which led to subsequent FDA approval for treatment of R/R B cell malignancies, including high-grade B cell lymphomas, DLBCL and transformed FL [141,142]. Hence, it is currently being investigated in other subtypes of B cell NHL including FL, MCL and MZL [9]. CD19-directed CAR T cells have also proven to be effective in R/R CLL patients [143].

Due to their immunomodulatory function, and in particular their potentiating effects on effector T cells, IMiDs could further improve response rates with CAR-T therapy in B cell NHL patients. In vitro and in vivo preclinical data strongly support the rationale for combining CAR-T cells with IMiDs such as lenalidomide, however, there are serious considerations regarding the management of expected toxicities [140]. Considering that both CAR-T therapy and IMiDs can lead to severe neutropenias and cytokine release syndrome, such combinations could lead to significant toxicity and will require careful design of future clinical trials. Interestingly, the CAR-T therapy JCAR017 is currently being investigated in combination with checkpoint blockade or CELMoDs including avadomide to determine the safety and efficacy of such combinations (NCT03310619). In addition, axicabtagene ciloleucel is being investigated in a phase II trial in combination with lenalidomide in aggressive/relapsed NHL [141].

**Table 1 ijms-22-08572-t001:** Select published phase II/III clinical trials with IMiDs and CELMoDs in CLL and B cell NHL.

Study/Year	Phase	No. of Patients	Arms	ORR (%)	CR (%)	PR (%)	PFS (mo)
CLL first line							
Badoux et al., 2011 [14] Follow up, 2013 [144]	II	60 35 LTRs/60	Len mono	65	15 71	50 29	NA NA
Chen et al., 2011 [145] Follow up 2014 [146]	II	25	Len mono	56 72	0 20	56 52	NA 40.4
James et al., 2014 [109]	II	40 29	Len + R (med age 56) Len + R (med age 70)	95 79	20 10	75 69	19 20
Jones et al., 2016 [147]	II	49	Len + Prevnar-13	75	2	73	NA
Chen et al., 2019 [148]	II	31	Len + Dexa	74	10	64	27
CLL Relapsed/Refractory							
Chanan-Khan et al., 2006 [73]	II	45	Len mono	47	9	38	NA
Ferrajoli et al., 2008 [69]	II	44	Len mono	32	7	25	NA
Wendtner et al., 2016 [149]	II	104	Len mono	42	8	34	22.3
Vitale et al., 2016 [110]	II	34	Len + O	71	24	47	16
Costa et al., 2015 [150]	II	21	Len + O	48	0	48	NA
Badoux et al., 2013 [151]	II	59	Len + R	66	12	54	17.4
Chavez et al., 2016 [152]	II	25	Len + R	46	0	46	14
Foa et al., 2016 [153]	III	160 154	Len mono PBO	NA NA	9.4 NA	47.5 NA	33.9 9.2
Strati et al., 2016 [154]	II	120	(LEN + R) Treatment naïve: Relapsed:	73 64	35 28	38 36	TTF 22 34
B-cell NHL first line							
Fowler et al., 2014 [12]	II	50 FL,30 MZL, 30 SLL	Len + R	90	63	27	53.8
Ruan et al., 2015 [103]	II	36 MCL	Len + R	92	64	28	NA
Martin et al., 2017 [101]	II	51 FL	Len + R	95	72	23	60
Morschhauser et al., 2018 [102]	II	513 FL	Len + R	61	48	13	36
Zucca et al., 2019 [155]	II	77 FL	Len + R	81	36	45	NA
Becnel et al., 2019 [156]	II	27 MZL	Len + R	93	70	22	59.8
Vitolo et al., 2014 [106]	II	45 DLBCL	Len + R-CHOP	92	86	6	NA
Nowakowski et al., 2015 [11] update 2021	II	60 DLBCL	Len + R-CHOP	97	73	24	NA
Tilly et al., 2018 [107]	II	80 FL	Len + R-CHOP	94	74	20	NA
B-cell NHL Relapsed/Refractory							
Witzig et al., 2011 [95]	II	108 DLBCL, 57 MCL, 19 FL, 33 TL	Len mono	35	13	22	3.7
Wiernick et al., 2008 [94]	II	26 DLBCL,5 FL, 15 MCL, 3 TL	Len mono	35	NA	NA	4
Eve at al., 2012 [100]	II	26 MCL	Len mono	31	8	23	14.6
Habermann et al., 2009 [96]	II	15 MCL	Len mono	53	20	33	5.6
Ferreri et al., 2016 [157]	II	46 DLBCL	Len mono	NA	NA	NA	1-year PFS: 70%
Tuscano et al., 2014 [158]	II	22 FL, 3 MZL, 3 SLL	Len + R	74	44	30	12.4
Chong et al., 2015 [159]	II	30 FL, 14 MCL, 4 SLL, 2 MZL	Len + R	62.8	NA	NA	22.2
Zinzani et al., 2013 [160]	II	23 DLBLC	Len + R	35	NA	NA	NA
Wang et al., 2013 [105]	II	32 DLBCL, 4 FL, 9 TL	Len + R	33	13	20	3.7
Morschhauser et al., 2016 [161]	II	71 DLBCL, 13 MCL	Len + Ob	30.6	16.5	14.1	NA
Thieblemont et al., 2016 [162] update	III	325 DLCBL 325 CLBCL	Len mono PBO	NA NA	21 14	NA NA	NA 68
Jerkeman et al., 2016 [108] update	II	50 MCL	Len + R+I	83	41	41	NA
Wang et al., 2016 [163]		30 MCL	Len mono	27	13	14	NA
Andorsky et al., 2016 [164]	III	61 FL, 16 MZL, 13 MCL	Len + R	56	20	36	NA
Leonard et al., 2019 [19]	III	147 FL, 31 MZL	Len + R	78	34	44	39.4
Avadomide clinical trials							
Carpio et al., 2020 [66]	I	97 DLBCL	Ava mono	28	9	19	2
Rasco et al., 2019 [24]	I	5 NHL	Ava mono	60	20	40	NA
Michot et al., 2020 [22]	Ib	19 DLBCL, 53 FL, 1 MZL	Ava + Ob	68	34	34	16

Abbreviations: CHOP, cyclophosphamide, doxorubicin, oncovin and prednisone; CLL, chronic lymphocytic leukemia; CR, complete response; DLBCL, diffuse large B cell lymphoma; Dexa, dexamethasone; FL, follicular lymphoma; I, Ibrutinib; Len, lenalidomide; LTR, long-term responders; MCL, mantle cell lymphoma; MZL, marginal zone lymphoma; NA, Not available; O, ofatumumab; Ob, obinutuzumab; ORR, overall response rate; PBO, placebo; PFS, progression free survival; PR, partial response; R, rituximab; SLL, small lymphocytic lymphoma; TL, transformed lymphoma; TTF, time to failure.

## 8. Conclusions

IMiDs and CELMoDs are rapidly emerging as a favorable chemotherapy-free treatment option partner for the treatment of B cell malignancies due to their potent immunomodulatory effects on diverse cellular components that reside within lymphoma TMEs, in particular T cells and NK cells. Future advances in our understanding regarding the MOA of IMiDs and CELMoDs, as well as studies revealing potential synergies with other novel immunotherapies such as BsAbs/BiTEs, immune checkpoint inhibitors and CAR-T therapy, should help to establish new therapeutic strategies in the clinic for CLL and B cell NHL patients, with the aim of establishing long-term disease control or disease eradication.

## Figures and Tables

**Figure 1 ijms-22-08572-f001:**
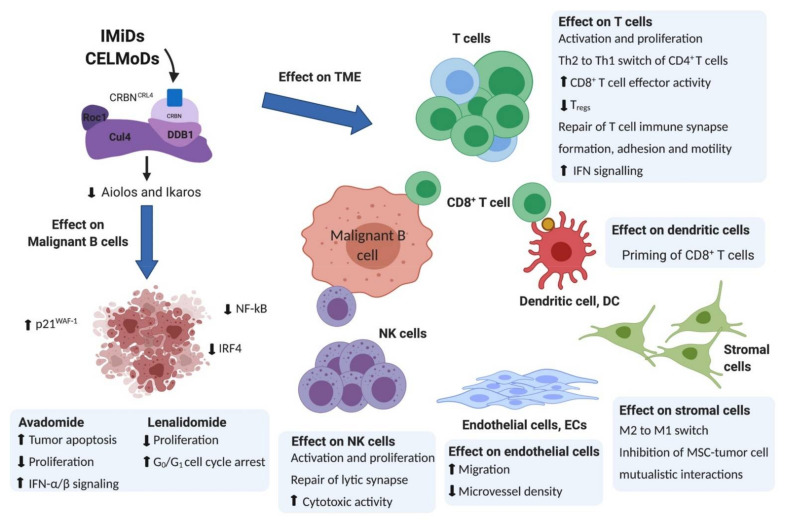
Mechanism of action of IMiD lenalidomide and CELMoD avadomide and their pleiotropic effects on various cell populations of the TME in hematological B cell malignancies including CLL and B cell NHL. Figure was created with BioRender.com.

**Figure 2 ijms-22-08572-f002:**
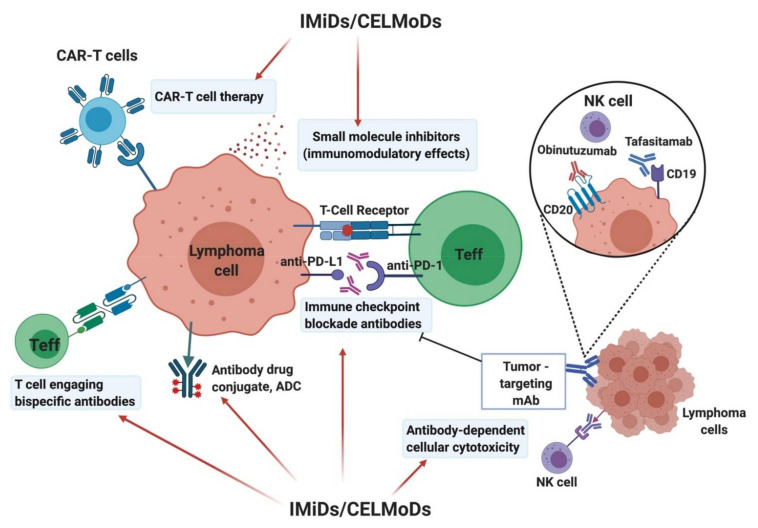
Current select immunotherapies where IMiDs/CELMoDs have the potential to provide complementary therapeutic effects and enhance the generation and persistence of anti-tumor immune responses when used in combination. Figure was created with BioRender.com.

## Data Availability

Not applicable.

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
