# Peer review of "Immunomodulatory Drugs for the Treatment of B Cell Malignancies"

_ijms, 2021, doi:10.3390/ijms22168572_

Round 1

Reviewer 1 Report

Minor comments

  1. [129] ‘lenalidomide has been shown higher efficacy against ABC-DLBCL cells compared to GCB-DLBCL’
    there’s no any comment about ABC, GCB-subtype even abbreviation
    it would be better to explain about ABC, GCB-subtype in diffuse large B-cell lymphoma
  2. Overall focused on lenalidomide. This is insufficient explain for CELMoDs(avadomide) and how they are different.
  3. It was difficult to fully understand the benefits of combination therapy.
    Emphasize the necessary of combination therapy and limitation of using as a single-agent.
  4. Immune checkpoint blockade means inhibitor
    [248] ‘the expression of immune checkpoint blockade molecule PD-1 on NK cells‘ change to immune checkpoint or exhaustion marker
  5. Typing error
    [184] regulatory T cells (Tregs)T

Author Response

Response to reviewer 1:

We are very grateful for the positive review and helpful comments and suggestions that we have now incorporated in our revised manuscript attached (tracked changes throughout). We have made the follow corrections to the minor suggestions recommended by reviewer 1.

Minor comments

  1. [129] ‘lenalidomide has been shown higher efficacy against ABC-DLBCL cells compared to GCB-DLBCL’
    there’s no any comment about ABC, GCB-subtype even abbreviation
    it would be better to explain about ABC, GCB-subtype in diffuse large B-cell lymphoma. RESPONSE - We have now defined these subtypes throughout the paper, and thank the reviewer for this suggestion that has improved understanding of the topic and review.
  2. Overall focused on lenalidomide. This is insufficient explain for CELMoDs(avadomide) and how they are different. RESPONSE - We have now added additional details within our Review on the CELMoDs and Avadomide that we hope provides sufficient detail. The MOA of Avadomide is currently being studied and we anticipate that results will be published in due course that will help explain the different substrates that are altered by CRBN targeting and modulation of its substrate recognition (altering the E3 ligase complex that modulates expression of critical transcription factors). We have focussed on the ability of CELMoDs and avadomide to trigger potent type I and II IFN signaling in patient T cells (Blood 2021) - superior immunostimulatory properties compared to Lenalidomide.
  3. It was difficult to fully understand the benefits of combination therapy.
    Emphasize the necessary of combination therapy and limitation of using as a single-agent. RESPONSE: We thank the reviewer for this suggestion and have added additional context text to aid reader understanding on the need to identify combination therapy for CLL and NHLs.
  4. Immune checkpoint blockade means inhibitor
    [248] ‘the expression of immune checkpoint blockade molecule PD-1 on NK cells‘ change to immune checkpoint or exhaustion marker. RESPONSE: done
  5. Typing error
    [184] regulatory T cells (Tregs)T. RESPONSE: done

Reviewer 2 Report

The review by Ioannou et al. provides a comprehensive overview of the current state of the use of immunomodulatory drugs for the treatment of B cell malignancies with a major focus on the effects of lenalidomide and avadomide. Overall, the manuscript is well written and provides a strong summary of the current state of the field.

Comments:

  1. Figures 1 and 3 are low resolution and some of the text should be enlarged. Please replace with high resolution figures.
  2. General text editing is needed.

Author Response

We thanks the Reviewer for their kind words and are grateful for the improvements and suggestions made that we have now addressed -

new Figures with enlarged text and general text corrections made throughout.

The review by Ioannou et al. provides a comprehensive overview of the current state of the use of immunomodulatory drugs for the treatment of B cell malignancies with a major focus on the effects of lenalidomide and avadomide. Overall, the manuscript is well written and provides a strong summary of the current state of the field.

Comments:

  1. Figures 1 and 3 are low resolution and some of the text should be enlarged. Please replace with high resolution figures.
  2. General text editing is needed.